# A Machine Learning Approach to Model Interdependencies between Dynamic Response and Crack Propagation

**DOI:** 10.3390/s20236847

**Published:** 2020-11-30

**Authors:** Thomas Fleet, Khangamlung Kamei, Feiyang He, Muhammad A. Khan, Kamran A. Khan, Andrew Starr

**Affiliations:** 1Through-Life Engineering Services, Cranfield University, Bedford MK43 0AL, UK; t.fleet@cranfield.ac.uk (T.F.); k.kamei@cranfield.ac.uk (K.K.); feiyang.he@cranfield.ac.uk (F.H.); a.starr@cranfield.ac.uk (A.S.); 2Aerospace Engineering Department, Khalifa University, Abu Dhabi PO Box 127788, UAE; kamran.khan@ku.ac.ae

**Keywords:** machine learning, thermomechanical fatigue, fatigue crack growth, damage detection

## Abstract

Accurate damage detection in engineering structures is a critical part of structural health monitoring. A variety of non-destructive inspection methods has been employed to detect the presence and severity of the damage. In this research, machine learning (ML) algorithms are used to assess the dynamic response of the system. It can predict the damage severity, damage location, and fundamental behaviour of the system. Fatigue damage data of aluminium and ABS under coupled mechanical loads at different temperatures are used to train the model. The model shows that natural frequency and temperature appear to be the most important predictive features for aluminium. It appears to be dominated by natural frequency and tip amplitude for ABS. The results also show that the position of the crack along the specimen appears to be of little importance for either material, allowing simultaneous prediction of location and damage severity.

## 1. Introduction

Coupled thermo-mechanical analyses are important on structures subjected to combined thermal and mechanical loads. A large number of high-performance engineering structures, such as components in internal combustion engines including reciprocating pistons, turbine blades, turbo impellers, exhaust, and bypass valves, are designed to operate under complex thermo-mechanical loads throughout their service life [1,2,3,4]. The complexity in the analysis is associated with an increase in a plastic strain where metallic structures are loaded at high temperatures (e.g., creep and creep-fatigue). This increase in plasticity dramatically complicates the understanding of crack growth and thermo-mechanical fatigue performance, as the fundamental linear elastic fracture mechanic (LEFM) assumption of a highly localized crack-tip plastic zone can be violated under these load conditions.

Therefore, the fast and accurate inspection of the severity of the damage in the component under thermo-mechanical loads is a critical and very challenging task. A variety of non-destructive techniques are used in the industry to inspect structural assets for damage detection and remedial action. A great deal of encouraging research is also taking place around using simple vibration tests and measurements of structural dynamic properties to infer the correct location and severity of damage in engineering structures [5,6,7,8,9,10,11,12,13,14,15,16]. The dynamic response of a structure can be used to measure the elastic properties of the constituent material [17]. Often, a combination of modal parameters like natural frequency and tip amplitude is used to model the relationship between thermomechanical loads and crack propagation, and they showed good correlation [18,19]. However, adequate predictive capability of fatigue crack propagation under complex thermomechanical loads remains a challenge for the field. 

In recent years, the rapid advances in computational power and the development of powerful statistical learning algorithms have led to the increased adoption of machine learning (ML) techniques in a variety of industries [20]. ML techniques have huge potential to predict the interdependencies and relationships of complex problems which cannot be easily described using conventional modeling techniques. Due to a large number of variables, the fatigue crack propagation under complex loads and in heterogeneous materials, failure mechanism classification, fault detection, etc. are all well-suited problems for machine learning. A large amount of research has applied ML techniques to identify the structural crack severity. Some of them proposed ML based methods for crack identification in beams and plates [21,22,23]. Liu and Zhang [24] employed convolutional neural networks (CNN) for crack damage detection in thin aluminum plates. Kourehli et al. [25] presented a method to identify the breathing cracks in Timoshenko beam under single or multiple moving mass. They applied an online sequential extreme ML algorithm as an inverse problem to predict crack depths and locations. Some research applied ML techniques in actual applications. Rageh et al. [26] proposed an automated damage detection framework to detect stiffness degradation that results from the initiation and growth of fatigue cracks in stringer-to-floor beam connection of the riveted steel railway bridges. The proposed method relies on Proper Orthogonal Decomposition (POD) and Artificial Neural Networks (ANNs) to identify damage location and intensity under non-stationary, unknown train loads. Wu et al. [27] proposed a crack detection approach in rotating shafts based on acoustic emission (AE) signals and ML. The research used Multiple Linear Regression (MLR), Artificial Neural Networks (ANN) and Adaptive Neural-Fuzzy Inference System (ANFIS) methods to investigate the relationship between crack intensity and domain features. Similarly, Gecgel et al. [28] proposed a simulation-driven ML framework to estimate the crack size in a spur gear pair using simulated vibrations signals. The results showed that the Decision Tree Classifier (DTC) performed best in the estimation of small crack sizes. Hasan and Kim [29] employed a combination of Fast Fourier Transform (FFT) analysis, a genetic algorithm signal pre-processing stage, and a k-Nearest Neighbours (KNN) classification model to determine the health of a spherical carbon steel tank structure. Apart from the crack identification in these metal components, some research also used recursive partitioning and artificial neural networks (ANN; deep learning frameworks) to predict the crack rating of pavements. They employed some explanatory variables such as the average daily traffic and truck factor, roadway functional class, asphalt thickness, and pavement condition time series data in the model formulation [30]. 

On the other hand, a great amount of research also utilised ML techniques to predict crack propagation rate or fatigue life of structure [31]. Karvelis et al. [32] used a deep learning neural network to detect the complex corrosion-enhanced fatigue of ship hulls. The model was able to detect the presence of a corrosion pit induced crack with over 99% accuracy with a t-Distributed Stochastic Neighbour Embedding (t-SNE) dimensionality reduction algorithm as a pre-processing step. Yang et al. [33] successfully demonstrated the application of Support Vector Machine (SVM) models to reduce the dimensionality of complex stress concentration factors using the linear kernel method. Similarly, Lu et al. [34] used a combination of machine learning algorithms including SVM, Neural Networks and Random Forests in combination with FEA and stochastic load simulation to accurately predict the remaining fatigue life of slender coastal bridges under complex traffic, wind and wave loads. Machine learning has also been used in a more abstract theoretical context. For example, Guilleminot and Dolbow [35] utilized MCMC (Monte Carlo Markov Chain) analysis to model stochastic fracture propagation paths materials with heterogeneous microstructure from base principles. Nezhad et al. [36] used dimensionality reduction algorithms to adequately simplify complex, nonlinear fracture mechanisms in heterogeneous materials, feeding the principal components into an optimization algorithm to determine the most probable fracture patterns.

While a variety of similar work exists, very few were found where the predictive feature importance was evaluated. Much of the research focussed solely on the satisfactory prediction of the existence or the severity of induced structural damage with little information given to the understanding of how or why the models predicted the way they did. Some of this can be attributed to the model choices made by the researcher; some models cannot be adequately interpreted or understood. In general, the more complex the model, the more powerful it is, and the more likely it is to be able to describe very complex problems adequately. However, as complexity increases, the less able one is to adequately explain how or why it is making its predictions. Therefore, based on the existing experimental data, this paper proposed a Ridge regularised multiple linear regression model that is capable of predicting the severity (and possibly location) of crack damage using simple dynamic response data, with an emphasis on the modal parameters interpretability. By evaluating the model coefficients, one could infer an understanding of the underlying importance of each of the measured properties and could also understand how these features change with material and boundary conditions. 

## 2. Materials and Methods

### 2.1. Specimen Parameters and Experimental Data Collection

Two representative materials, 2024 sheet Aluminium and Fused Deposition Modelling (FDM) 3D printed ABS, were chosen from previous research. Various specimens were manufactured with the geometry shown in Figure 1a.

The experimental data were taken from previous experiments [12,37]. The experimental set-up is shown in Figure 1b. The specimen was fixed on the shaker and heated to several temperatures. The shaker provided the mechanical loads. Impact tests were carried out to determine the fundamental frequency of the specimen, and it was measured by a laser vibrometer. Then, the vibration test was performed at the fundamental frequency. In the case of the crack propagation, the displacement amplitude of the beam tip stays reduced. Therefore, the shaker was stopped to measure the new frequency. The impact tests were performed again to find the new fundamental frequency. The new lower frequency was set to the shaker. Thus, the maximum amplitude can be achieved again while vibrating. The new fundamental frequency was maintained until the next amplitude drop. This procedure was repeated until the catastrophic failure of the specimen for propagating crack specimens.

The research performed experiments on both aluminium and ABS. Figure 1c shows the evolution of crack propagation in aluminium. Figure 1d shows the example of typical crack propagation path in 3D printed ABS. The collected experimental data consisted of crack position (mm), temperature (°C), natural frequency (Hz), tip amplitude (mm), and crack depth (mm) for each observation. 

Previously, the empirical model was developed between the crack depth/location and structural dynamic response, but there were a large number of coefficients in this model, and it was a high order which led to the model being difficult to interpret. Therefore, a suitable and more concise model is required, which can not only predict the crack propagation accurately but also can provide insight on the physical meanings of the coefficients. A Ridge-regularised multiple linear regression model was chosen to meet these requirements.

### 2.2. Analysis Setup

All the data analysis detailed in this paper was conducted directly from the raw experimental data from [12,37]. All of the work described in this document was conducted using the Python programming language [26] and various data science packages and libraries. All data processing and manipulation were performed using the pandas [38] and NumPy [39] libraries. The machine learning work was conducted using the scikit-learn library [40], and all graphs and visualisations were produced using the Matplotlib, Seaborn, Chartify and Altair libraries [41,42,43]. It is shown as a flow chart in Figure 2.

### 2.3. Linear Regression Modelling

The primary focus of this work is not simply to generate accurate predictions but to use the machine learning model to inform the theory, an interpretable model was a key factor. Therefore, a family of machine learning models, called “Generalised Linear Models”, were investigated. Because these models broadly function in a similar way, it means their coefficients, and thus their internal workings, are easily introspectable in terms of the original data that trained them. A Ridge regularised multiple linear regression model, as a member of the Generalised Linear Model family, was used because of its interpretability. Equation (1) shows the format of the model where $\hat{y}$ is the predicted crack depth, $\theta_{i}$ is the model coefficient for the $x_{i}$ feature (crack location, temperature, natural frequency and tip amplitude)—alternatively, in matrix form where θ is the coefficient matrix and x is the matrix of features, as shown in Equation (2). Because these coefficients are intrinsically linked to the features, they can be read directly to infer the importance and relative impact of each feature on the dependent variable:(1)$$\hat{y}=\theta_{0}+\theta_{1}x_{1}+\theta_{2}x_{2}+\ldots +\theta_{n}x_{n}$$
(2)$$\hat{y}=\theta \cdot x$$

The coefficient matrix θ is adjusted by an optimisation algorithm, typically a version of the gradient descent algorithm. A closed form solution called the Normal Equation, which means an iterative optimisation algorithm is not always required is applied to minimise the errors between model predictions and actual data. This error is calculated by a cost function, which is commonly the sum of the squared errors with an extra term as shown in Equation (3) performing the ‘regularisation’ where $h_{\theta }$ is the hypothesis function such that $MSE\left(X,\;h_{\theta }\right)$ is the mean squared error of a linear regression hypothesis $h_{\theta }$ on a training dataset X. $\theta^{T}$ is the transposed coefficient matrix and $x^{\left(i\right)}$, $y^{\left(i\right)}$ are the ith values for the feature matrix and actual crack depth, respectively, α is a hyperparameter that controls the degree of regularisation. The model is trained by this adjusting and optimising process:(3)$$MSE\left(X,\;h_{\theta }\right)=\frac{1}{m}\overset{m}{\underset{i=1}{\sum }}{\left(\theta^{T}x^{\left(i\right)}-y^{\left(i\right)}\right)}^{2}+\alpha \frac{1}{2}\overset{m}{\underset{i=1}{\sum }}\theta_{i}^{2}$$

The regularisation term $\alpha \frac{1}{2}\sum_{i=1}^{m}\theta_{i}^{2}$ places additional constraints on the values of the coefficients, as it forces the learning algorithm not only to fit the data but encourages it to limit the absolute value of the coefficients as much as possible whilst still maintaining accuracy in order to avoid the ‘overfitting’ problem common in machine learning where the model optimises its coefficients to the training data so much that it cannot perform well on new, previously unseen data.

The regularised linear model is highly sensitive to the scale of the input features. For comparing the model coefficients directly and properly, a typical pre-processing step, standardisation of the data, was performed and can be described by Equation (4), where $z_{i}$ is the standardised value, μ is the mean, and $\sigma_{i}$ is the standard deviation, corresponding to each $x_{i}$ feature. This operation effectively subtracts the mean and scales data to unit variance, which reduces all the features down to a comparable scale while keeping the shape and distribution of the data intact. It also helps model examining the underlying effects revealed by the model carefully:(4)$$z_{i}=\;\frac{\left(x_{i}-\mu_{i}\right)}{\sigma_{i}}$$

### 2.4. Machine Learning Modelling

Because of the different material behaviour, and to allow proper comparison of the coefficients, two identical but independent regression models were trained: one for Aluminium, and one for ABS.

The data were split blindly and randomly into training and test sets with an 80/20 split. Model hyperparameters, Equation (3) were adjusted until the model reaches satisfactory performance. This practice ensured that the model was always evaluated on previously unseen data, and the evaluated performance was therefore more representative of a real-world predictive problem.

During model training, it was essential to determine the performance of the model without violating the “test on unseen data” principle. This was accomplished by implementing a K-fold cross validation methodology. The K-fold cross-validation method used in this research is shown in Figure 3. The model was used to predict values for the entire dataset once the final one obtained good performance on the K-fold cross-validated training score and the reserved test set.

## 3. Results and Discussion

### 3.1. Training Data

Determining the structural response is vital in engineering applications. In this study, the experimental data of natural frequency and structural amplitude are considered. It is observed that the natural frequency of ABS and Aluminium show some similar patterns. The natural frequency keeps decreasing in response to an increasing crack depth and increasing temperature as shown in Figure 4. There is a nonlinear reduction in natural frequency with increasing crack depth. The impact of temperature effect on natural frequency is far more apparent in the Aluminium than in the ABS despite the ABS being tested at values up to 66% of its Tg. This is likely due to the higher elastic modulus and the relative consistency of isotropic sheet metal properties compared to the additive layer manufactured ABS.

For the aluminium samples, increasing crack depth and temperature leads to a higher tip amplitude. This is the expected effect of a reduced natural frequency and can be easily rationalised to the detrimental effect of a crack in the specimen. The ABS, however, exhibits the reverse behaviour; as the crack depth increases, the natural frequency drops as in the aluminium, but the tip amplitude decreases rather than increases as shown in Figure 5. This can be explained by considering the temperatures tested during the experiments. The maximum temperature for the aluminium specimens is 200 °C, well below the temperature at which any meaningful material transformations may occur during the short duration of the test. The maximum temperature tested in the ABS samples is 70 °C, which is close to the glass transition temperature. This counterintuitive amplitude response is the proximity of the higher temperature tests to the glass transition temperature of the material as well as the effect of stress crazing as discussed before. A more complete view of the relationship between amplitude and frequency in the experimental data is shown in Figure 6.

### 3.2. Machine Learning Model

The predicted and the actual crack depth for the entire dataset are shown in Figure 7. Because of the different material behaviour discussed earlier and to allow proper comparison, two identical but independent models were trained: one for Aluminium and one for ABS. Figure 8 shows the model predictions plotted against the actual crack depth measurements. In general, the predictive accuracy is encouraging for both materials with a root mean squared error (RMSE) of 0.176 mm for Aluminium and 0.256 mm for ABS. In simple terms, these metrics can be interpreted as the average predictive error of the models. Table 1 summarises the performance metrics.

The performance of the ABS model is slightly lower than that of the aluminium model. This is most likely due to the higher variance and less isotropic nature of the additive layer manufactured ABS and subtle interactions between a propagating crack and the material layers. This is best indicated in the lower R^2^ value corresponding to a lesser degree of variance by the model compared to the aluminium data. Because the models were trained on standardised data, the coefficients can be directly compared and evaluated to determine feature importance. In the case of standardising data, the absolute value of the feature coefficient is a reasonable proxy for relative importance. These data are presented in Figure 8 with the actual coefficient values in Figure 9.

Natural frequency and temperature appear to be the most important predictive features for Aluminium, while ABS appears to be dominated by natural frequency and amplitude. Most interestingly, the position of the crack along the specimen appears to be of little importance for either material. This is slightly counterintuitive as theory would suggest the closer the crack to the fixed position of the specimen, the more pronounced the effect on dynamic response. This can be explained by considering that the effect of crack position would show up in the natural frequency and tip amplitude terms, and therefore would be captured by the model under those coefficients, leaving the crack position coefficient small and therefore relatively unimportant.

The result also shows the opposite polarity of the amplitude coefficient for the two materials. The positive coefficient for Aluminium indicates that an increase in crack depth increases amplitude with all elements held constant. However, the negative coefficient for ABS implies the exact reverse. The importance of coefficients in order is given in Table 2. The model clearly shows that, whilst using all of the features as predictors yields the lowest RMSE, the removal of amplitude and crack location does not significantly increase the error. The removal of temperature has a much more significant effect on the error, increasing it by approx. 47% relative to the all features baseline. By far the most significant effect is seen by the removal of natural frequency from the feature pool, which increases the error by nearly 200%. This is a strong indication that the natural frequency is the dominant feature in the prediction of crack depth, and its effect on the model error was greater than the rest of the features combined.

## 4. Conclusions

Experimental data gathered from previous works [12,37] were analysed and explored using machine learning techniques to accurately predict damage severity based on the dynamic response of the specimen during a simple oscillation test under various temperatures. It was found that the Ridge Regularised multiple linear regression model adequately described the system and was capable of generating sufficiently accurate predictions, whilst maintaining the interpretability and introspection common to generalised linear models. It was able to predict crack depth to within an RMSE of 0.176 mm and 0.256 mm for Aluminium and ABS, respectively, and, even when the crack position was excluded from the feature pool, it still maintained reasonable accuracy, indicating a good early potential for a damage detection methodology.

Furthermore, the introspection of the model revealed interesting findings of the physical system. The natural frequency was found to be by far the most influential feature for the prediction of damage severity—this fact held for both materials despite their significant differences in material properties and manufacturing methods. By inspecting the fitted model and evaluating the differences in coefficient values for the two materials, it was possible to infer whether coefficients may be boundary condition or material specific. For example, the amplitude coefficient seems likely to be material (or manufacturing method) specific as the test conditions between the two were identical, yet this coefficient had significantly different values (opposite signs). This coefficient seems likely to be related to the materials elastic modulus as this would explain such a large difference between the two materials tested.

Another potential material-specific coefficient corresponds to the temperature. In reality, this coefficient may link to the materials’ thermal softening profile. Therefore, it is ultimately the underlying atomic structure and bonding mechanisms. The remaining coefficient for the crack position was found not to differ significantly between the two materials. It can also be explained by analytical modelling. Because the relative crack locations and experimental setup were the same for both materials, this means that the boundary conditions and structural continuity are constant in the model. This had a relatively unimportant impact on the crack depth prediction for both materials.

The coefficient of natural frequency and temperature appears as important predictive features for Aluminium, and frequency and amplitude for ABS. The crack location appears to be of little importance for both materials. Nevertheless, this might be against the theory which would suggest that the closer the crack to the fixed position, the more pronounced the effect on dynamic response. However, as the crack location is not defined through the theoretical models, in future studies, the training of the data need to be performed using the theoretical models so that more rigorous ML models can be obtained.

## Figures and Tables

**Figure 1 sensors-20-06847-f001:**
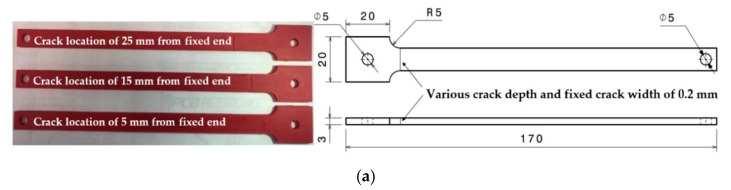
(**a**) specimen geometry; (**b**) experimental set-up; (**c**) crack propagation path in aluminium; (**d**) evolution of crack propagation in FDM ABS [12,37].

**Figure 2 sensors-20-06847-f002:**
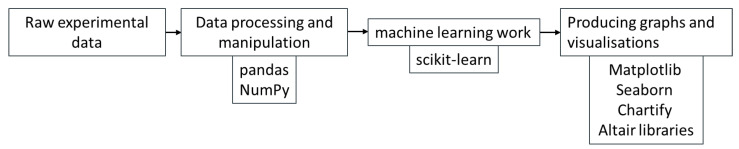
Flow chart of analysis steps.

**Figure 3 sensors-20-06847-f003:**
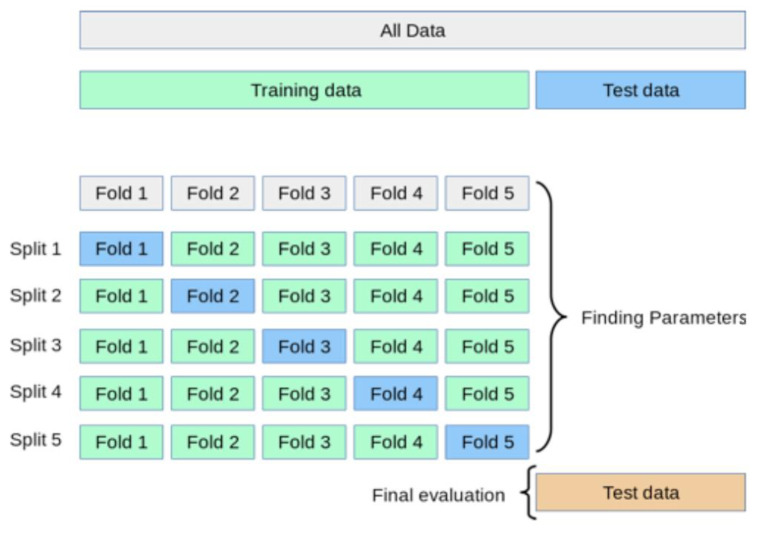
Schematic showing a K-fold cross-validation.

**Figure 4 sensors-20-06847-f004:**
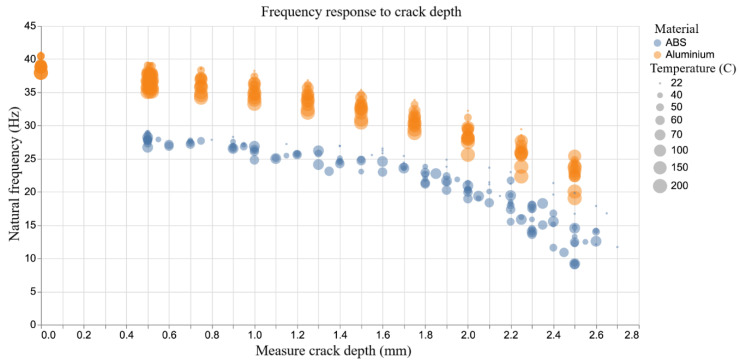
Natural frequency response to crack depth.

**Figure 5 sensors-20-06847-f005:**
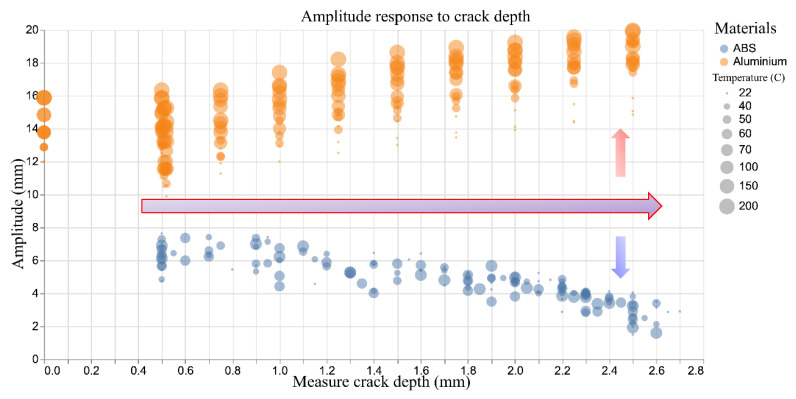
Tip amplitude response to crack depth.

**Figure 6 sensors-20-06847-f006:**
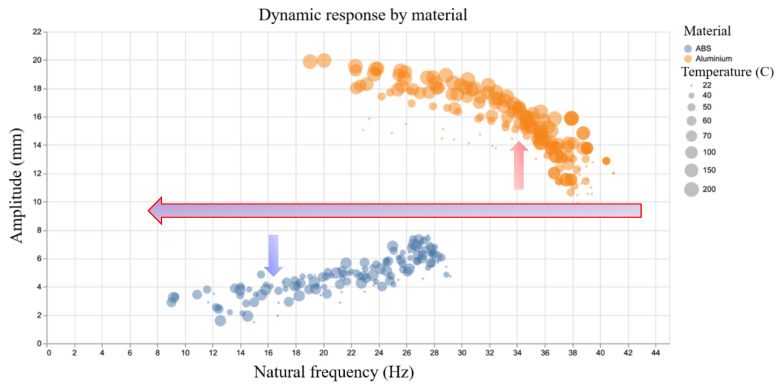
Dynamic response by material.

**Figure 7 sensors-20-06847-f007:**
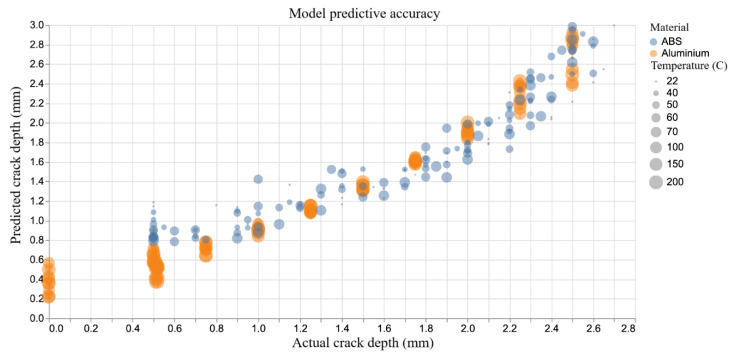
Model predictive accuracy.

**Figure 8 sensors-20-06847-f008:**
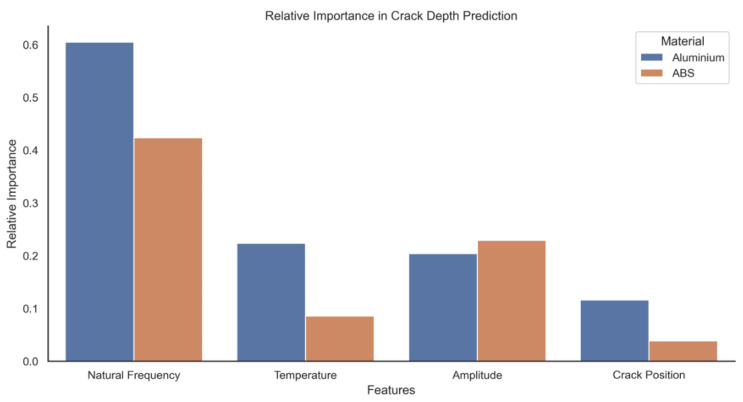
Relative feature importance derived from model coefficients (scaled).

**Figure 9 sensors-20-06847-f009:**
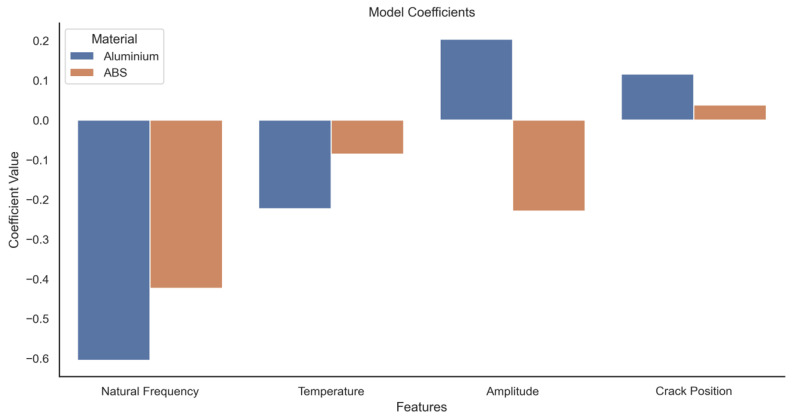
Model coefficient values (scaled).

**Table 1 sensors-20-06847-t001:** Model accuracy scores.

Metric	Aluminium	ABS
RMSE	0.176 mm	0.256 mm
R^2^	0.95	0.86

**Table 2 sensors-20-06847-t002:** Model coefficient values (scaled).

Feature	Aluminium Coefficient	ABS Coefficient
Natural Frequency	−0.636304	−0.338966
Temperature	−0.244674	−0.092730
Amplitude	0.161097	−0.281892
Crack Position	0.119964	0.081306

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
