# Peer review of "A Machine Learning Approach to Model Interdependencies between Dynamic Response and Crack Propagation"

_sensors, 2020, doi:10.3390/s20236847_

Round 1
Reviewer 1 Report
The paper is interesting and well prepared.
I propose to change describtion x axis in Figures 4,5 and 7 - in my opinion better is to shown relative crack depth than crack depth.
Also the literature should be improved, because the machine learning for crack identification is the subject of research interest from many years.
Reviewer 2 Report
This paper developed a machine learning approach to model interdependencies between thermomechanical loads and crack propagation. The paper needs some improvements, as following:
- More recent related references should be added and cited in the main text.
- More information about the experimental testing should be added.
- More contents about the machine learning model should be added.
- The section of Conclusion should focus on the innovation results of the paper.
Reviewer 3 Report
The findings, using a generalized linear model to describe the crack growth from dynamic response are relatebale and the method presented is a good way to show how data science techniques can, in theory, be employed to discover new relations for exisiting phenomena.
However the main findings (crack growth ~ nat. frequency) are not supprising, it is still a relevant example for the use of such models in materials testing. It would be nice, if the performance of the model can be compared to a simple linear fit based on the natural frequency, or even better an analytical description derived from beam theory.
The biggest flaw in the paper however is the choice of the title and the motivation. However it is true that describing crack growth under thermomechanical loading is a big headache in engineering, this study does not adress that phenomenon. Thermomechanical loads are arising from changing temperature profiles, leading to thermalliy induced stresses ect. The connected feature of thermo-mechanical fatigue and subsequent crack growth require the changes of the mechanical- and the thermal load to be roughly within the same cycle time scale. This is even the first sentence in the Wikipedia article on this. The pheonmenon studied here is crack growth under pure isothermal fatigue.
Employing the method presentend under TMF conditions would lead to much more experimental difficulties. It is therefore highly recommended to change the title of the paper, to not mislead potential readers here.
Further comments are included in the attached .pdf file.

Round 2
Reviewer 2 Report
The authors have revised the manuscript based on the reviewer's comments. The paper can be accepted in the present form.